# Transformer-based Working Memory for Multiagent Reinforcement Learning with Action Parsing

**Yaodong Yang**[1], **Guangyong Chen**[2,3,*] **Weixun Wang**[4], **Xiaotian Hao**[4], **Jianye Hao**[4], **Pheng Ann Heng**[1,5]

[1]Department of Computer Science and Engineering, The Chinese University of Hong Kong
[2]Zhejiang Lab [3]Zhejiang University [4]Tianjin University
[5]Institute of Medical Intelligence and XR, The Chinese University of Hong Kong
yydapple@gmail.com, gychen@zhejianglab.com,
{wxwang,xiaotianhao,jianye.hao}@tju.edu.cn, pheng@cse.cuhk.edu.hk

## Abstract

Learning in real-world multiagent tasks is challenging due to the usual partial observability of each agent. Previous efforts alleviate the partial observability by historical hidden states with Recurrent Neural Networks, however, they do not consider the multiagent characters that either the multiagent observation consists of a number of object entities or the action space shows clear entity interactions. To tackle these issues, we propose the Agent Transformer Memory (ATM) network with a transformer-based memory. First, ATM utilizes the transformer to enable the unified processing of the factored environmental entities and memory. Inspired by the human's working memory process where a limited capacity of information temporarily held in mind can effectively guide the decision-making, ATM updates its fixed-capacity memory with the working memory updating schema. Second, as agents' each action has its particular interaction entities in the environment, ATM parses the action space to introduce this action's semantic inductive bias by binding each action with its specified involving entity to predict the state-action value or logit. Extensive experiments on the challenging SMAC and Level-Based Foraging environments validate that ATM could boost existing multiagent RL algorithms with impressive learning acceleration and performance improvement.

## 1 Introduction

Multiagent reinforcement learning (MARL) is a way of learning how to make sequential decisions when multiple agents interact with the environment given the constraints of partial observability [25]. There are many representative real-world applications of MARL including wireless network optimization [7], autonomous driving [5] and energy distribution [29]. Partial observability has been a long-standing challenge for reinforcement learning (RL) since the real environment state can be partially hidden from the agent [2], and this issue becomes even worse for MARL as the involving dynamics are more complex than single-agent RL with teammates concurrently exploring.

Typically, to tackle the partial observability, most previous efforts propose to summarize the historical observation trajectories to provide additional information for the current local observation. For instance, in a navigation task, the only way to distinguish two T-junctions that look identical is to remember the past key observations before entering into either T-junction [2]. Among them, gated Recurrent Neural Networks (RNNs) [33] serve as the most popular mechanism to represent memory for RL [19, 10, 13] and so does for MARL [26, 31]. Given the superior performance achieved by the transformer on sequential modeling tasks over RNNs [34], some pioneer efforts have been

---

[*]Corresponding author.

36th Conference on Neural Information Processing Systems (NeurIPS 2022).

contributed to replace RNNs with transformer as the memory structure in single-agent RL, resulting in superior performance especially on tasks that require the long horizon reasoning ability with sequential memory [17, 28, 3]. However, the transformer-based methods remain limited because the computational complexity increases quadratically with the length of historical observation trajectories. Meanwhile, these RNN-based and transformer-based works seldom consider the multiagent system characters that the multiagent observation consists of a number of object entities and the action space contains meaningful and clear entity interactions. In this paper, we make the first attempt to leverage the transformer as a memory processing structure to facilitate MARL algorithms under the partially observable setting through carefully considering the above multiagent system characters.

Our first contribution is to introduce the transformer-based working memory [17, 23] into MARL. For the first time, we develop a unique transformer-based multiagent memory structure with the help of the working memory updating mechanism by explicitly considering the allied agents for the factorized multiagent observation space. Specifically, we dynamically maintain a fixed-capacity memory for each agent and update it with the imitated human working memory process where a small amount of information is stored in mind while working with it for effective decision making [22]. Furthermore, considering the factored multiagent environment, we leverage the transformer to generate new memory by uniformly processing the sequential memory slots and observed environment entities. Additionally, with the fixed-capacity memory, the computational complexity of the transformer keeps the same when agents continuously interact within the environment.

Our second contribution is to introduce an action semantic inductive bias and implement it with a technique named Entity-Bound Action Layer. The main insight behind the Entity-Bound Action Layer is that each action in the action space interacts with particular involved environment entities according to its semantic meaning [36]. For example, in one battle scenario, an attack action mainly causes impacts on the targeted specific enemy. Thus, we are motivated to parse the action space to explicitly model the interaction, so that each action is uniquely bound to its targeted entity to predict the state-action value or logit. Entity-Bound Action Layer brings lots of benefits, including increase the computational flexibility of the policy network and making the agent's action more explainable.

Furthermore, we find that the above contributions can be seamlessly integrated into a compact model, what we term as Agent Transformer Memory (ATM) network, with simultaneously processing the sequential working memory and the spatial entities, and binding unique entity embeddings to actions by the action semantic inductive bias. Extensive experiments on the challenging SMAC and Level-Based Foraging environments validate that ATM could be easily plugged in many existing MARL algorithms with impressive learning acceleration and performance improvement across various tasks.

## 2 Background

### 2.1 Markov Games

Markov games are a multi-agent extension of Markov Decision Processes [16]. They are defined by a state transition function, $T : S \times A^1 \times ... \times A^N \to P(S)$, which defines the probability distribution over all possible next states, $P(S)$, given the current global state $S$ and the action $A^i$ produced by the $i$-th agent. Note that the reward is usually given based on the global state and actions of all agents $R^i : S \times A^1 \times ... \times A^N \to \mathbb{R}$. If all agents receive the same rewards, i.e. $R^1 = ... = R^N$, Markov games are fully-cooperative: a best-interest action of one agent is also a best-interest action of others [18].

We use the partially observable Markov games as our settings, in which each agent $i$ receives a local observation $o^i : Z(S, i) \to O^i$. Thus, each agent learns a policy $\pi^i : O^i \to P(A^i)$, which maps each agent's observation to a distribution over its action set, to maximize its expected discounted returns, $J^i(\pi^i) = \mathbb{E}_{a^1 \sim \pi^1, ..., a^N \sim \pi^N, s \sim T}[\sum_{t=0}^{\infty} \gamma^t r_t^i(s_t, a_t^1, ..., a_t^N)]$ with $\gamma \in [0, 1]$ as the discounted factor. To help address the partial observability issue, we record the historical experience of agent $i$'s action-observation $\tau^i$ to provide extra information. Traditionally, an RNN such as a GRU is used in the agent's individual policy or Q-value function network for abstracting its historical experience [30].

### 2.2 Transformer

Harnessing the transformer to handle the partial observability has been investigated in RL [32, 17, 28]. The core of the transformer is the self-attention mechanism [34]. The self-attention adopts three matrices, $\mathbf{Q} = T_{in}\mathbf{W}^q, \mathbf{K} = T_{in}\mathbf{W}^k, \mathbf{V} = T_{in}\mathbf{W}^v$ representing a set of queries, keys and values

respectively where $T_{in} \in \mathbb{R}^{n_e \times d_e}$ is the input of transformer and $\mathbf{W}^q \in \mathbb{R}^{d_e \times d_k}$, $\mathbf{W}^k \in \mathbb{R}^{d_e \times d_k}$ and $\mathbf{W}^v \in \mathbb{R}^{d_e \times d_v}$ are learnable weight matrices. $n_e$ is the number of input entities while $d_e$, $d_k$, $d_v$ denote the dimensions of input entities, keys (or queries) and values. The attention output is computed as

$$Att(\mathbf{Q}, \mathbf{K}, \mathbf{V}) = softmax(\frac{\mathbf{Q}\mathbf{K}^T}{\sqrt{d_k}})\mathbf{V}, \tag{1}$$

where $d_k$ also behaves as a scaling factor. In practice, a multi-head structure is employed to allow the model to focus on different representation sub-spaces. The multi-head attention in transformer uses $H$ different sets of learned projections as

$$\begin{aligned} T_{mha} &= MHA(\mathbf{Q}, \mathbf{K}, \mathbf{V}) = concat(head_1, ..., head_H)\mathbf{W}^{MHA}, \\ where\ head_i &= Att(\mathbf{Q}, \mathbf{K}, \mathbf{V}; \mathbf{W}_i^q, \mathbf{W}_i^k, \mathbf{W}_i^v). \end{aligned} \tag{2}$$

$\mathbf{W}^{MHA}$ is the learnable matrix for the concatenated vectors from each attention head. After the multi-head self-attention, two Add & Norm operations with residual layers are applied to compute the transformer's output $T_{out}$.

$$\begin{aligned} T_{res} &= LN(T_{mha} + T_{in}), \\ T_{out} &= LN(T_{res} + ReLU(Linear(T_{res}))), \end{aligned} \tag{3}$$

where $LN$ is the layer normalization and $ReLU$ represents the ReLU activation function. $Linear$ is a linear layer. The transformer block outputs $T_{out}$ with inputting $T_{in}$.

## 3 Agent Transformer Memory Network

In this section, we give a detailed description of the ATM framework as shown in Figure 1. To handle the partial observability, ATM provides each agent with a slot-based memory to store the past key information in $\tau^i$, which is updated by the working memory mechanism. To restrict the computational complexity of applying the transformer on the whole trajectories, ATM uses a fixed-capacity memory buffer $\mathbf{M}$. Next, ATM embeds all the spatial entities and memory slots with relative positional embeddings as $T_{in}$ and inputs them into the transformer block. Using spatio-temporal embeddings $T_{out}$ outputted from transformer, ATM calculates individual Q-values or policy logits with Entity-Bound Action Layer by semantically binding unique entity embeddings in $T_{out}$ to actions.

### 3.1 Agent Transformer Memory

Here we give the detailed design of the ATM network with the compartmentalized slot-based memory mechanism. Formally, each agent maintains its own memory buffer matrix $\mathbf{M} \in \mathbf{R}^{n_M \times d_M}$. Each row of $\mathbf{M}$ is a memory slot $m^j$ with size $d_M$ where $j \in [1, 2, ..., n_M]$, which stores the agent's memory on the past timesteps. There are fixed $n_M$ slots in $\mathbf{M}$ and the memory buffer is updated at each timestep.

When applying the transformer to handle the various observation space, we take advantage of the rich-entity character of the multiagent environment. As shown in Figure 1's left part, we divide entities in the agent observation space into three spatial entity sets: $o_{self}^i$, $\mathbf{o}_{ally}^i$ and $\mathbf{o}_{ent}^i$. The self entity $o_{self}^i$ contains agent $i$'s individual features and is embedded as $e_{self}^i$ via an embedding layer. Similarly, the entities of other allied agents $\mathbf{o}_{ally}^i$ contain agent $i$'s observation to each other agents and are embedded as $\mathbf{e}_{ally}^i$ via another embedding layer. Then the remaining entities (such as enemies or environmental objects) $\mathbf{o}_{ent}^i$ are embedded as $\mathbf{e}_{ent}^i$ via another embedding layer. Besides feeding these spatial entities into the transformer block, to enable the agent memory, we also regard agent $i$'s sequential memory slots $m^{i,1}, ..., m^{i,n_M}$ as memory entities and input them into the transformer block after being embedded into $\mathbf{e}_m^i$ via an embedding layer. With the embedded sequential entities from memory and spatial entities from the local observation, the transformer block's input is built as

$$T_{in}^i = [e_{self}^i; \mathbf{e}_{ally}^i; \mathbf{e}_{ent}^i; \mathbf{e}_m^i]. \tag{4}$$

After the transformer block updating all entity embeddings, the resulting outputting embeddings are

$$T_{out}^i = TRM(T_{in}^i) = [e_{self}^{i,out}; \mathbf{e}_{ally}^{i,out}; \mathbf{e}_{ent}^{i,out}; \mathbf{e}_m^{i,out}], \tag{5}$$

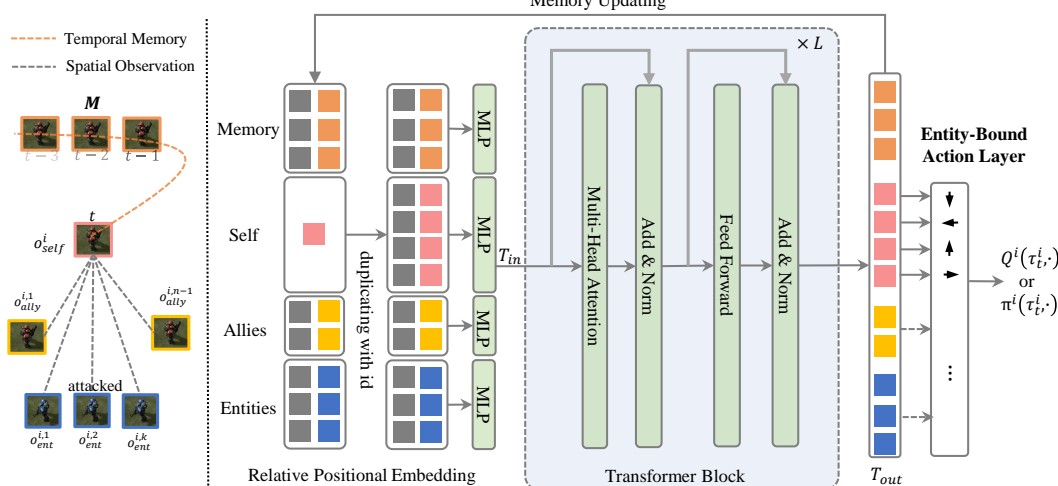

Figure 1: Agent Transformer Memory Network. The left part illustrates an example of an agent's local observation with the sequential memory $\mathbf{M}$ and spatial entities including itself $o^i_{self}$, allies $\mathbf{o}^i_{ally}$ and enemies $\mathbf{o}^i_{ent}$. The right part is the network architecture of ATM. All memory slots and observed entities are first encoded with relative sequential or spatial positions (grey squares) and then embedded via a multi-layer perceptron (MLP). For memory or duplicated self entities, the grey square is a one-hot ID. For allies or other entities (such as enemies), the grey square is a relative distance between the central self entity and the allies or other entities. Next, all entity embeddings denoted as $T_{in}$ are inputted into the transformer block. Then the outputting entity embeddings $T_{out}$ are used to compute the Q-values or action probabilities with the Entity-Bound Action Layer. Meanwhile, the working memory mechanism continuously upgrades the agent's memory with the transformer's output at each timestep. "$\times L$" means that there are $L$ layers for the transformer block.

where $TRM$ indicates the standard transformer block of Eq. (1), (2) and (3). Then each agent's $T_{out}$ is used to update its memory buffer $\mathbf{M}$ and compute the state-action value or policy.

As the transformer processes these embedded memory and spatial entities as a set, the spatial or sequential order information of entities would be lost if we do not encode it into the entity features. Therefore, we treat the agent's self entity as the central entity and encode the relative positional embeddings into all other entities related with it. For the sequential memory entities, we perform the one-hot embedding $\mathbf{M} = [\mathbf{M}, \mathbf{I}]$ where $\mathbf{I}$ is the identity matrix to indicate the passing timesteps compared with the current timestep $t$. For the spatial entities, we encode the relative distances (i.e., the x-coordinate distance and y-coordinate distance) with the central self entity into the entity features. Next, we elaborate the working memory updating schema with $T_{out}$ of each agent.

## 3.2 Working Memory Updating Schema

To facilitate agents to hold past information, we imitate the working memory process in cognitive science, which describes a cognitive system with a limited capacity that can hold information temporarily [21, 22]. Holding information in mind and manipulating it, such a theoretical concept has been utilized in single-agent deep reinforcement learning to alleviate the partial observability problem and has shown to surprisingly increase the reasoning ability of agents [17, 23]. In detail, working memory drops the oldest memory slot $m^{n_M}_t$ and add the newest memory slot updated from the transformer block's output. The new memory $\mathbf{M}_{t+1}$ is updated as

$$
\begin{aligned}
m_{t+1} &= tanh(h_t W_M + b_M), \\
\mathbf{M}_{t+1} &= [m_{t+1}; \mathbf{M}_t[:-1]],
\end{aligned}
\tag{6}
$$

where $h_t = T_{out}[0;]$ is the self entity embedding outputted by the transformer and $tanh$ is the tanh activation. $W_M$ is a weight matrix and $b_M$ is a bias vector. The working memory mechanism behaves as a first-in-first-out queue to manage the agent's memory of the past timesteps.

There are two benefits of ATM compared with RNN-based networks. First, ATM effectively utilizes the factorized structure existing in the agent observation space and employs the powerful transformer block to handle these various meaningful entities to extract internal concurrent relations such as coordination of teammates. Second, ATM's multi-slot memory mechanism could provide the short-cut recurrence to focus on longer sequential information, replacing a gated RNN's single path of information flow with a network of shorter self-attention paths [17].

### 3.3 Parsing Action by Entity-Bound Action Layer

After ATM outputting $T_{out}$, we need to calculate individual state-action values $Q^i$ or the action probabilities of the agent's policy $\pi^i$. One common way is using linear layers to map the self entity embedding $h = T_{out}[0;]$ to all action nodes' values [17] (similar for the policy probability logits) of the output layer such as

$$Q^i = Q^i(h, a), \forall a \in A. \tag{7}$$

However, mapping the self entity to all actions does not effectively utilize the aggregated spatio-temporal embeddings from transformer as different actions may require different information. When designing the action space, each action is given its own semantic meaning to interact with different involved entities in the environment. Such an inductive bias of actions naturally requires to bind the involved relevant entity embedding from $T_{out}$ to the semantically corresponding actions such that

$$Q^i = Q^i(h_a, a),$$
$$\text{where } h_a \in T_{out} \text{ and } h_{a_b} \neq h_{a_c} \text{ if } b \neq c. \tag{8}$$

To realize Eq. (8), we propose the Entity-Bound Action Layer to uniquely bind the outputting entity embeddings in $T_{out}$ with actions. For each action, we map the most relevant entity embedding to it by parsing the action (e.g. mapping the enemy entity embedding in $T_{out}$ to the corresponding attacking action in $A$). Then we use the bound entity embedding to compute each action's state-action value or probability logit via linear layers. When computing, all used entity embeddings $h_a \in T_{out}$ share the same linear layers. If one entity maps to more than one action, we replicate this entity with one-hot ID in $T_{in}$ to make sure that each action has one unique outputting entity embedding in $T_{out}$. Figure 1 shows an example of applying the Entity-Bound Action Layer on four moving actions related to the agent itself. The corresponding self entity in $T_{in}$ is duplicated four times with ID to make sure that each moving action could receive its unique entity embedding from $T_{out}$. Then the uniquely bound entity embeddings in $T_{out}$ are used to calculate the final state-action values or probabilities via the shared linear layers. Using the unique entity embeddings to calculate the corresponding state-action values or probabilities allows each action to flexibly focus on the different inputting entities such as the memory slots or the allies via self-attention, thus improving the computational flexibility compared with using the same outputting embedding $h = T_{out}[0;]$ to compute all actions' values.

The Entity-Bound Action Layer is based on ATM's factorized observation space and the compartmentalized memory slots, which can not be trivially implemented with RNNs on concatenated observations. For the working memory updating schema in Section 3.2, if the self entity $e_t^{i,s}$ is duplicated $n_{self}$ times in $T_{in}$ to map different actions, the new memory slot is computed by averaging all the output embeddings originated from the self entity. Then the memory is updated as

$$h_t = mean(T_{out}[0:n_{self}]),$$
$$m_{t+1} = tanh(h_t W_M + b_M),$$
$$\mathbf{M}_{t+1} = [m_{t+1}; \mathbf{M}_t[:-1]]. \tag{9}$$

## 4 Experiment

### 4.1 StarCraft Multi-Agent Challenge

In this section, we evaluate our method[2] in the StarCraft II decentralized micromanagement tasks and use StarCraft Multi-Agent Challenge (SMAC) environment [31] as our testbed, which has become a commonly-used benchmark for evaluating state-of-the-art MARL approaches. At the beginning of each episode, the enemy units are going to attack the allies. We train multiple agents to control

---

[2]Code is available at `https://github.com/CNDOTA/NeurIPS22-ATM`.

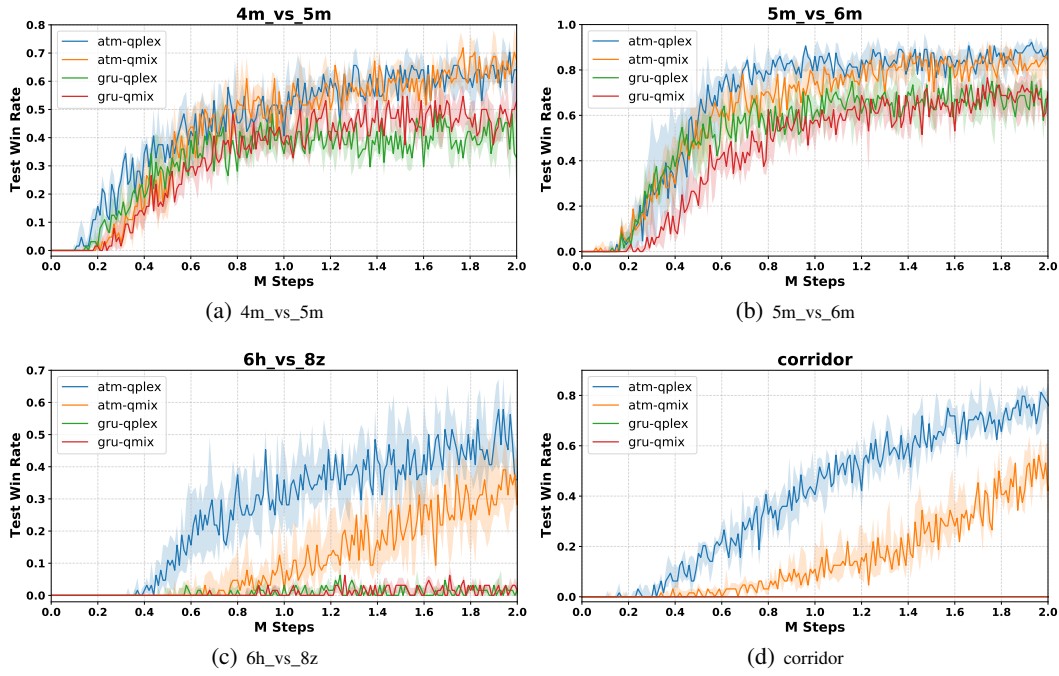

Figure 2: Results on different StarCraft Multi-Agent Challenge scenarios.

allied units respectively to beat the enemy, while a built-in handcrafted AI controls the enemy units. Training and evaluation schedules such as the testing episode number and training hyper-parameters are kept unchanged. The version of StarCraft II is 4.6.2. We build ATM based on the pymarl [31] and ATM is used as the agent network when plugged into MARL algorithms.

First, we describe the features of each entity type. In the SMAC scenarios, the exact observation of agent $i$ is a set of entities $e$ where $e \in \{o^i_{self}, o^i_{ally}, o^i_{enemy}\}$ and $d(e, i) < d^i_{sight}$ while $d(e, i)$ is the distance between the entity $e$ and agent $i$, and $d^i_{sight}$ is agent's sight range. The self entity $o^i_{self}$ contains each agent's own health feature, movable features, and the possible type and shield features. Each ally entity $o^i_{ally}$ contains features such as the health, the distance, relative $x$ coordinate, and relative $y$ coordinate between this ally and the agent itself. The other entities here are the enemies, which have similar features as the ally entities. The meta-information such as the agent id and last action is added into the agent's self entity. Agents cannot share information such as their first-person observations among themselves as we assume there is no communication. At the same time, if one entity is in both agent $i$ and agent $j$'s sight ranges, then agents $i$ and agent $j$'s own local observations include the same seen entity. The discrete set of actions that agents are allowed to take consists of move[direction], attack[enemy id], stop and no-op. Here we assign actions of move[direction], stop and no-op to the self entity while actions of attack[enemy id] to corresponding enemy entities. For the memory part, we use three memory slots and four attention heads with each head having 16 hidden dimensions. Then the embedding layers are of 64 hidden dimensions. For mapping the entity embeddings in $T_{out}$ to state-action values, we use one linear layer shared by all actions' bound entity embeddings and set the output dimension to 1. More details are provided in the Appendix.

We test ATM with QMIX [30] and QPLEX [35], two of the most representative algorithms on SMAC. We compare ATM with GRU, which is widely adopted in MARL algorithms in pymarl [31] and is validated to obtain consistent advantages over MLP in SMAC [30]. We perform the experiments on four maps: 4m_vs_5m, 5m_vs_6m, 6h_vs_8z, and corridor. The last two maps are almost the most difficult maps where previous methods fail to learn [31]. Results are averaged over 6 independent training runs with different random seeds and the resulting plots include the median performance as well as the 25-75% percentiles. As shown in Figure 2, we can see that ATM consistently improves the performance of QMIX and QPLEX on these maps. Surprisingly, ATM makes the breakthrough on 6h_vs_8z and corridor where GRU-based QMIX and QPLEX fail. The learned strategies with ATM

exhibit the delicate micro-manipulation. For example, in corridor, agents stand in a circle formation to maximize their attacking damage while avoiding being attacked from behind.

### 4.1.1 Ablation Study of ATM

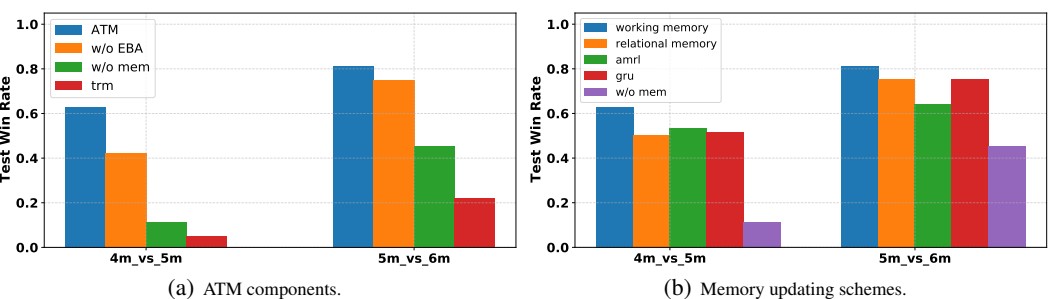

(a) ATM components.  (b) Memory updating schemes.

Figure 3: Ablation Study of ATM.

We also conducted the ablation study of each part of ATM, which is shown in Figure 3(a). All the results are averaged over 6 independent training runs with different random seeds and we show the median test win rate. The 'w/o mem' is ATM without any memory on QMIX by setting the memory slot number to 0. The 'w/o EBA' is ATM without Entity-Bound Action Layer on QMIX, which instead uses linear layers to map the self entity embedding to all action nodes' values. The 'trm' means that ATM degenerates to the basic transformer without memory or Entity-Bound Action Layer on QMIX. We could see that, on this map, the memory contributes a lot to the algorithm performance while the Entity-Bound Action Layer is also an essential technique for improving the performance. If without the two key parts of ATM, the model degenerates to the basic transformer and gets the lowest test win rate, which validates the effectiveness of each component of ATM.

Meanwhile, we study the effect of the memory updating schema. Besides the working memory, we introduce another two state-of-the-art memory updating schemes from the single-agent RL (details in Section A.1.2). The first one is relational memory [32], which is first proposed for tasks that require complex sequential reasoning ability. The most important idea of relational memory is to allow memory slots to interact with each other to perform complex relational reasoning with the information they remember. The second one is AMRL [4] which uses standard memory module to summarize short-term context, and then aggregate all prior states from the standard model without respect to order to provide advantages both in terms of gradient decay and signal-to-noise ratio over time. Figure 3(b) shows the results of these memory updating schemes. The working memory performs better than others while all memory updating methods substantially exceed the ATM without memory.

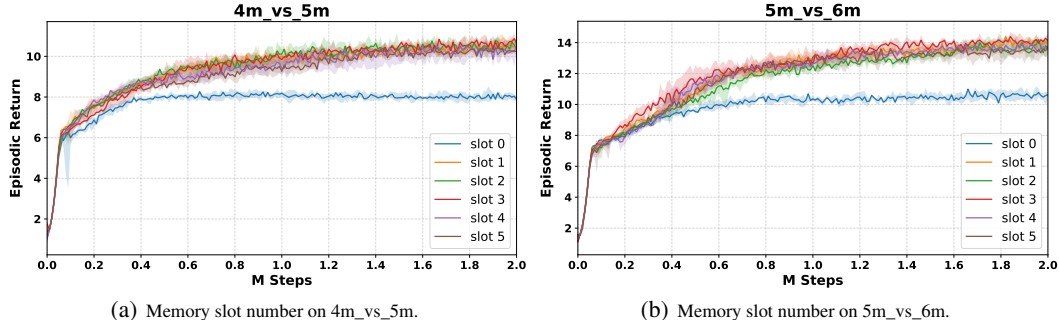

(a) Memory slot number on 4m_vs_5m.  (b) Memory slot number on 5m_vs_6m.

Figure 4: Memory slot number study.

### 4.1.2 Memory Slot Number

Next, we study the impacts of the memory slot number. We test ATM with memory slot numbers ranging from 0 to 5. The results are shown in Figure 4. We could see that when the number of

memory slots is larger than 0 which means with memory, ATM could obtain similar performance and achieves best when the slot number is 3. Indeed, if the number of the memory slots increases to the length of the past timesteps, ATM becomes a transformer over the whole sequential observation trajectories. But Figure 4 indicates that truncated observation trajectories such as 3 timesteps are enough while the computational complexity of the transformer increases quadratically with the length of the historical trajectory. We set slot number at 3 for all tasks in both SMAC and LBF.

### 4.1.3 Attention Illustration of ATM

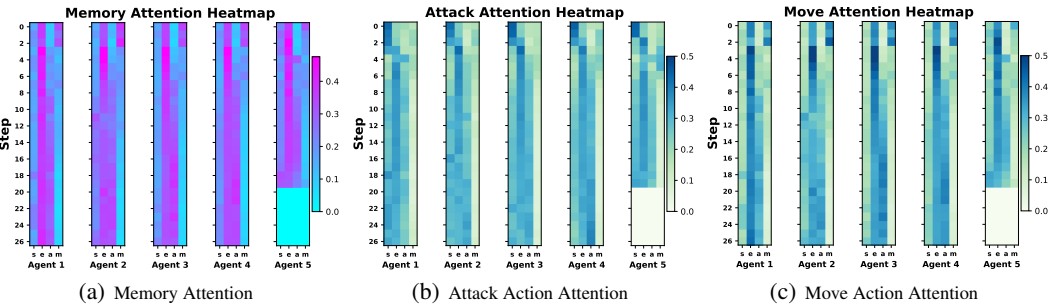

(a) Memory Attention      (b) Attack Action Attention      (c) Move Action Attention

Figure 5: Attention heatmaps, 's' is agent itself, 'e' are enemies, 'a' are allies, and 'm' is memory.

ATM utilizes the structure information in multiagent environment with self-attention, then the attention weights of entities could be illustrated to indicate the inner dynamics of ATM. Here we show how the new memory is generated by attending over entities and how the bound entities of actions attends over entities. As shown in Figure 5(a), the new generated memory at each timestep absorbs the enemy entities and ally entities most as the agent need to remember them. On the contrary, the self entity is less attended to be absorbed into memory as it is never lost in the sight range. Figure 5(b) indicates that the attack actions attend over the self, ally and enemy entities at the same time, which is consistent with the key strategy of focused fire. Meanwhile, Figure 5(c) attend most on enemies and allies, which means that the move actions also needs coordination. Besides, we also provide a more detailed object-oriented translation of the agent's decision process with entities in Appendix C.

## 4.2 Level-Based Foraging

We further test ATM on the classical Level-Based Foraging (LBF) tasks [1], where agents collect foods scattered randomly in a grid world. Agents and foods are assigned levels, such that a group of one or more agents can collect the food if the sum of their levels is greater or equal to the food's level. Agents can move in four directions, and have an action that attempts to load an adjacent food (the action will succeed depending on the levels of agents attempting to load the particular food). When one or more agents load a food, the food level is rewarded to the agents by the agent level. We test ATM on three distinct partially observable LBF tasks with variable agents and foods. For example, "15x15_3p_5f" means that the grid has a size 15x15 and there are 3 agents and 5 foods on the grid. The agent's observation space includes self entity, ally entities and food entities. We test on three challenging maps: "15x15_3p_5f", "15x15_4p_5f" and "15x15_4p_6f" and set the sight range of agents at 7 to introduce partial observability while ensuring that agents observe some object entities.

For the configuration of ATM, the memory slot number is 3 and 4 attention heads are used with each head having 16 hidden dimensions. The embedding layers for the agent, allies, and food entities are one-layer linear networks of 64 hidden dimensions. The action space of LBF is [None, North, South, West, East, Load] and we duplicate the self entity to map each action. To map the action entities in $T_{out}$ to state-action probability logits, we use a one-layer linear network shared by all action entities and set the hidden dimension to 1. Other hyper-parameters such as training and testing configurations are kept the same as the best parameters reported in the epymarl [27] after a grid search and could be referred to in Appendix. As MAPPO and MAA2C exhibit excellent performance on almost all tasks, we plug ATM into MAPPO and MAA2C to test ATM on the two on-policy algorithms while comparing with GRU-based and MLP-based ones. All the results are averaged over

5 independent runs with different random seeds for 5 million steps and the resulting plots include the median episodic return plus 25-75% percentiles while the highest episodic return is normalized to 1.

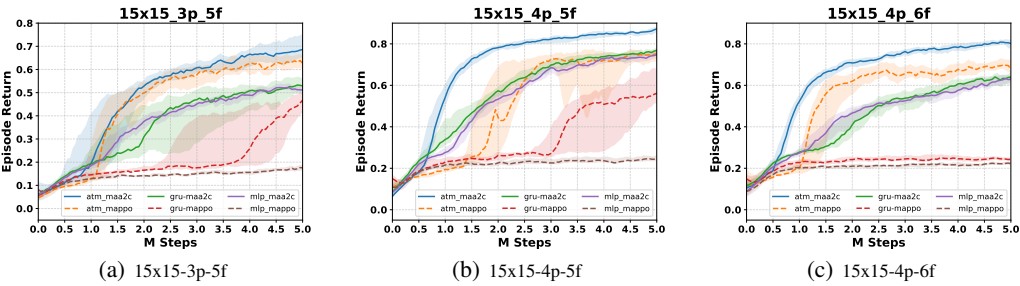

Figure 6: Results on different Level-Based Foraging scenarios with a sight range.

Figure 6 shows the results of the LBF tasks. We could see that, with the help of ATM, the learning speeds of MAPPO and MAA2C are improved by a large margin compared with GRU and MLP. Notably, GRU-based MAPPO and MLP-based MAPPO fail on the "15x15_4p_6f". However, with ATM, MAPPO easily learns from scratch. Meanwhile, ATM-MAA2C exhibits the best performance in all the scenarios. In the LBF environment where agents have a partial sight window, ATM exhibits superior learning speed and final performance when plugged into MAPPO and MAA2C.

## 5 Related Work

One of the most relevant classes of work is the single-agent RL algorithms with recurrent memory to solve the partially observable Markov decision process (POMDP). For deep reinforcement learning (DRL), DRQN [19] and RDPG [10] are the first two works to introduce RNNs into DRL to provide memory for agents. R2D2 [13] then studies the representational drift and recurrent state staleness phenomena while using LSTM in RL and proposes a "burn-in" updating method to better restore the initial state. At the same time, the attention mechanism is exploited to help update the memory. FRMQN [24] utilizes the key-value memory to perform the writing operation and the soft attention to perform reading operation with the spatio-temporal context vector. As an adapted version of FRMQN, DCRAC [23] maintains the separate external value and key memory blocks and uses the hidden state from the LSTM and query to aggregate relevant information from memory. Furthermore, RMC [32] employs the multi-head self-attention to let each memory attend over all of the other memories, and update its content based on the attended information to perform sequential relational reasoning. Next, AMRL [4] uses LSTM to summarize short-term context, and then aggregate all prior states over time from LSTM to provide advantages both in gradient decay and signal-to-noise ratio.

Some works are also explored to build agent memory with transformer. For example, WMG [17] uses the transformer to reason over a dynamic set of vectors representing observed and recurrent states in single-agent DRL. At the same time, GTrXL [28] found that the standard transformer architecture is difficult to optimize especially pronounced with RL objectives. Then they propose the modified GTrXL with a gating technique that substantially improves the stability and learning speed of the original transformer. Following GTrXL, CoBERL [3] and HCAM [14] use integrated or hierarchical memory structure to help organize the internal state with more than 10 millions of parameters. Besides online RL, transformers are also investigated in offline RL by treating RL as a sequence modeling problem such as Decision Transformer [6] and Trajectory Transformer [12].

On the other side, transformers are introduced into MARL for offline and transfer learning. Based on Decision Transformer, Multiagent Decision Transformer [20] studies the paradigm of offline pre-training with online fine-tuning in MARL on the SMAC platform. UPDeT [11] utilizes a transformer-based model to enable multiple tasks transferring in MARL through the transformer's strong generalization abilities. Unlike previous works in MARL using the transformer, here we focus on designing a transformer-based memory structure for agents with local observations to boost online MARL algorithms in the challenging partially observable environments. There are also some efforts focusing on the partial observability in the board multiagent domain. For instance, Subramanian et al. [8] extend the mean-filed method into the partially observable settings specially for the large-

scale multiagent system by maintaining a distribution over the mean action parameter. In [15], the SPARTA Monte Carlo search procedure is proposed to improve the agreed-upon policy for Hanabi, a cooperative partially observable card game. Besides, a model-based IPOMDP-net is designed for multi-agent planning under partial observability with an interactive belief update mechanism [9].

## 6    Conclusion

In this work, we propose the Agent Transformer Memory network to handle partial observability in MARL by specially considering the multiagent system characters. First, the transformer uniformly aggregates the agent's factorized observed entities and sequential memory slots to generate new internal memory and updates memory with the working memory updating schema. Second, we utilize Entity-Bound Action Layer to uniquely bind predefined entity with corresponding actions to compute the state-action values or probability logits. Experiments demonstrate ATM's excellent ability to speed up the learning and improve the performance of MARL algorithms on various tasks.

For future works, on the one hand, it is interesting to extend the idea of the Entity-Bound Action Layer to learn how to automatically map the ATM's outputting entity embeddings to the unique actions which are difficult to be configured manually or by simple heuristic rules. On the other hand, it is promising to introduce more advanced natural language processing techniques into the MARL framework to boost the performance of MARL algorithms by utilizing the rich semantic meanings which naturally exist in MARL environments.

## ACKNOWLEDGMENTS

The work was supported by National Key R&D Program of China (2022YFE0200700), National Natural Science Foundation of China (Project No. 62006219), Natural Science Foundation of Guangdong Province (2022A1515011579), and Hong Kong Innovation and Technology Fund under Project No. ITS/170/20 and Project No. GHP/080/20SZ.

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
