# A   Additional Materials on SMAC

## A.1   Configurations on SMAC

For all MARL algorithms we use in SMAC[3] [31] (under the MIT License), we keep the hyper-parameters the same as in pymarl[4] [31] (under the Apache License v2.0) except the ATM network. The ATM network is used as the individual Q-value network for agents and we here give the detailed network configurations of ATM in Table 1. To enable Entity-Bound Action Layer, we duplicate the self entity with ID in the transformer's input to actions of move[direction], stop and no-op while assign each enemy entity to corresponding actions of attack[enemy id]. When the transformer outputs the embeddings of these bound entities, we map the corresponding entity embeddings with the Entity-Bound Action Layer to unique action values or logits. Here we set the number of transformer blocks at 1 to make a fair comparison. The computation resources include NVIDIA Tesla V100 and AMD EPYC Rome.

Table 1: The network configurations of ATM on SMAC.

| Network Configurations | Value |
|---|---|
| self embedding layer number | 1 |
| self embedding layer hidden dimension | 64 |
| ally embedding layer number | 1 |
| ally embedding layer hidden dimension | 64 |
| enemy embedding layer number | 1 |
| enemy embedding layer hidden dimension | 64 |
| memory embedding layer number | 1 |
| memory embedding layer hidden dimension | 64 |
| memory slot number | 3 |
| query layer number | 1 |
| query layer hidden dimension | 64 |
| key layer number | 1 |
| key layer hidden dimension | 64 |
| value layer number | 1 |
| value layer hidden dimension | 64 |
| attention head number | 4 |
| attention head size | 16 |
| transformer block number | 1 |
| Entity-Bound Action Layer number | 1 |
| Entity-Bound Action Layer hidden dimension | 1 |

### A.1.1   ATM on Different Algorithms

Besides QMIX, we also test ATM on other classical algorithms such as IQL and VDN. Results are shown in Table 2 and Figure 7. Here we report both the median test win rate and mean episodic return with the 95% confidence interval. Results are averaged over 6 independent training runs with different random seeds. As we can see, ATM improves IQL and VDN when replacing GRU in the agent Q-value network, which validates the generality of ATM on different MARL algorithms.

Table 2: Results on Additional Different Algorithms.

| Method | GRU 4m_vs_5m | ATM 4m_vs_5m | GRU 5m_vs_6m | ATM 5m_vs_6m |
|---|---|---|---|---|
| IQL | 23.4% | 43.8% | 31.3% | 64.1% |
| VDN | 39.1% | 48.4% | 60.9% | 81.3% |

---

[3]https://github.com/oxwhirl/smac
[4]https://github.com/oxwhirl/pymarl

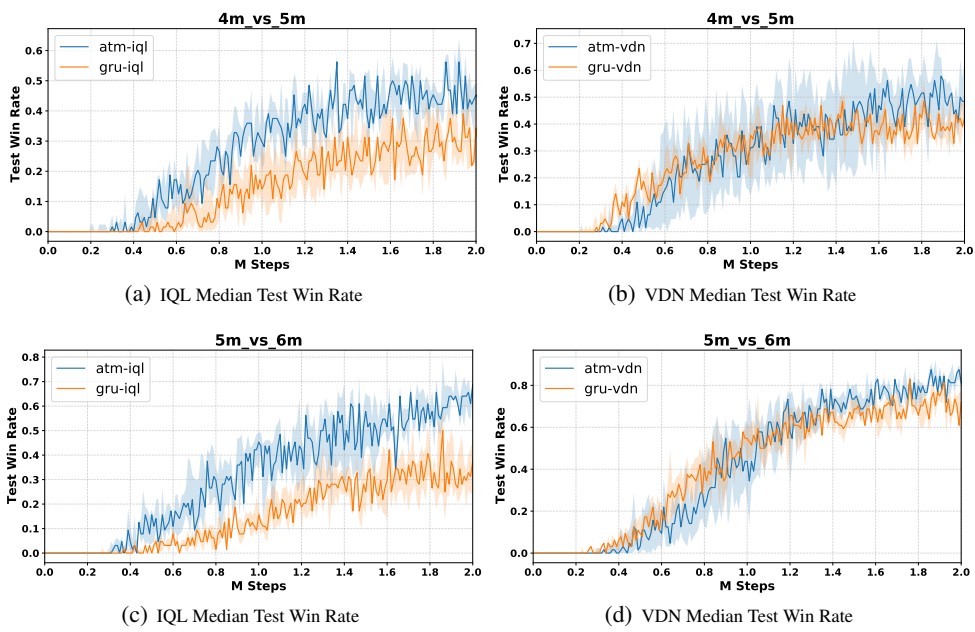

(a) IQL Median Test Win Rate

(b) VDN Median Test Win Rate

(c) IQL Median Test Win Rate

(d) VDN Median Test Win Rate

Figure 7: Results of IQL and VDN with ATM on SMAC.

### A.1.2 Details of Other Memory Mechanism

Here we give the details of the relational memory [32] and AMRL [4]. In the relational memory, each memory slot will attend over all of the other memories and input, and update its content based on the attended information. Here we incorporate this mechanism into our framework as the relational memory updating schema. The new memory $\mathbf{M}_{t+1}$ based on relational memory is updated as

$$\mathbf{M}_{t+1} = T_{out}[-n_M :], \tag{10}$$

where $T_{out}[-n_M :]$ indicates the last $n_M$ memory embeddings outputted by the transformer. Each outputting memory embedding in $T_{out}[-n_M :]$ has aggregated information over all the memory and observable entities. In the experiment, we also enhance the relational memory with the Entity-Bound Layer to improve its performance. On the other hand, AMRL first uses an LSTM to produce the hidden state $h_t$ from the observation embedding $e_t$:

$$\begin{aligned} e_t &= Linear(o_t), \\ h_t &= LSTM(e_t), \end{aligned} \tag{11}$$

where $Linear$ is the linear layers and $LSTM$ is the LSTM layer and here we also use the GRU version of LSTM in AMRL. Then AMRL uses a commutative aggregator function that combines all previous encodings $h_t$ in a time-independent manner:

$$m_t = g(m_{t-1}, h_t[:\frac{1}{2}]), \tag{12}$$

where $h_t[:\frac{1}{2}]$ denotes the first half of $h_t$, and $g$ denotes the aggregator function and we use the maximum operation as it consistently performs well in the original paper. Then the final action $a_t$ is produced by

$$\begin{aligned} c_t &= concat(h_t[\frac{1}{2}:], m_t), \\ a_t &= Linear(c_t). \end{aligned} \tag{13}$$

### A.2 Parameter Number

Here we give the individual Q-value network's parameter number when using ATM or GRU on different maps in Table 3.

Table 3: Parameter Numbers of Agent Network with ATM or GRU on SMAC.

|          | ATM   | GRU   |
|----------|-------|-------|
| 4m_vs_5m | 38.5k | 29.6k |
| 5m_vs_6m | 39.9k | 30.4k |
| 6h_vs_8z | 42.1k | 32.2k |
| corridor | 53.4k | 39.3k |

# B  Configurations on LBF

We follow the settings in epymarl[5] [27] (under the Apache License v2.0) and keep the hyperparameters the same as the hyperparameters of MAPPO and MAA2C with GRU or MLP found by a grid search as shown in Table 4. The computation resources include NVIDIA GeForce RTX 2080 and Intel Xeon CPU E5-2680 v4.

Table 4: Hyper-parameters of MAA2C and MAPPO on LBF.

|                        | MAPPO        | MAA2C        |
|------------------------|--------------|--------------|
| hidden dimension       | 128          | 128          |
| learning rate          | 0.0003       | 0.0005       |
| reward standardisation | False        | True         |
| network type           | MLP/GRU/ATM  | MLP/GRU/ATM  |
| entropy coefficient    | 0.001        | 0.01         |
| target update          | 0.01 (soft)  | 0.01 (soft)  |
| n-step                 | 5            | 10           |

ATM is used as the individual policy network for agents and we here give the detailed network configurations of ATM in Table 5. To enable Entity-Bound Action Layer, we duplicate the self entity with ID in the transformer's input to the actions of [None, North, South, West, East, Load]. When the transformer outputs these bound entity embeddings, we map each corresponding embedding with the Entity-Bound Action Layer to unique actions. Here we set the number of transformer blocks at 1.

Table 5: The network configurations of ATM on LBF.

| Network Configurations                      | Value |
|---------------------------------------------|-------|
| self embedding layer number                 | 1     |
| self embedding layer hidden dimension       | 64    |
| ally embedding layer number                 | 1     |
| ally embedding layer hidden dimension       | 64    |
| food embedding layer number                 | 1     |
| food embedding layer hidden dimension       | 64    |
| memory embedding layer number               | 1     |
| memory embedding layer hidden dimension     | 64    |
| memory slot number                          | 3     |
| query layer number                          | 1     |
| query layer hidden dimension                | 64    |
| key layer number                            | 1     |
| key layer hidden dimension                  | 64    |
| value layer number                          | 1     |
| value layer hidden dimension                | 64    |
| attention head number                       | 4     |
| attention head size                         | 16    |
| transformer block number                    | 1     |
| Entity-Bound Action Layer number            | 1     |
| Entity-Bound Action Layer hidden dimension  | 1     |

---

[5]https://github.com/uoe-agents/epymarl

## B.1 Parameter Number

Here we give the individual policy network's parameter numbers when using ATM, GRU or MLP on different scenarios in Table 6.

Table 6: Parameter Numbers of Agent Network with ATM, GRU or MLP on LBF.

|  | ATM | GRU | MLP |
|---|---|---|---|
| 15x15_3p_5f | 36.7k | 103.4k | 20.9k |
| 15x15_4p_5f | 37.4k | 103.9k | 21.4k |
| 15x15_4p_6f | 38.1k | 104.3k | 21.8k |

# C Translation of Agent's Decision Process with ATM

In this section, we show that ATM could somewhat reveal the black box of MARL decision process. We provide the translation of agent 0's decision process in one battle on 5m_vs_6m as shown in Table 7. Each row represents the entity which the action's bound entity focuses on. 'Move N' means moving north while 'Move S' means moving south. 'Move E' means moving east while 'Move W' means moving west. "Attack E0" means attacking enemy 0 and so on. The red text means that at this step, the agent chose the action of this column and focused on the entity as the text with the maximum attention weight. We could observe that, although four moving actions bind with the self entity, their focused entities are often different at the same step. It is consistent with the action semantic as different actions cause different effects on the involved parts of the environment. Furthermore, we could also see that this agent behaved based on its memory for many steps, which also validates the necessity of memory. Such a translation shows the decision process of the agent and provides the explanations of the agent's internal activities when computing the state-action values or logits.

Table 7: Translation of Agent 0's Decision Process during One Battle.

| Step | Stop | Move N | Move S | Move E | Move W | Attack E0 | Attack E1 | Attack E2 | Attack E3 | Attack E4 | Attack E5 |
|---|---|---|---|---|---|---|---|---|---|---|---|
| 0 | ally 3 | ally 3 | ally 3 | memory 0 | ally 3 | ally 1 | self | self | self | self | ally 1 |
| 1 | memory 1 | ally 2 | memory 0 | memory 2 | ally 2 | self | self | self | self | self | self |
| 2 | memory 0 | memory 0 | memory 0 | memory 0 | memory 0 | memory 0 | memory 0 | memory 0 | memory 0 | memory 0 | memory 0 |
| 3 | memory 1 | ally 0 | memory 1 | enemy 2 | ally 2 | memory 1 | memory 1 | enemy 2 | memory 1 | memory 1 | memory 1 |
| 4 | enemy 4 | ally 0 | enemy 2 | enemy 1 | ally 2 | enemy 4 | self | enemy 4 | self | enemy 4 | enemy 4 |
| 5 | ally 3 | ally 2 | enemy 2 | enemy 2 | ally 3 | enemy 4 | enemy 4 | memory 0 | enemy 4 | enemy 4 | enemy 4 |
| 6 | enemy 0 | ally 0 | enemy 0 | enemy 4 | ally 0 | ally 3 | enemy 1 | memory 0 | enemy 1 | memory 0 | enemy 4 |
| 7 | ally 0 | ally 0 | enemy 0 | enemy 4 | ally 0 | ally 3 | memory 0 | self | enemy 4 | memory 0 | enemy 4 |
| 8 | ally 0 | ally 0 | enemy 0 | enemy 2 | ally 0 | ally 3 | enemy 2 | ally 1 | enemy 3 | enemy 2 | enemy 1 |
| 9 | ally 0 | ally 0 | memory 0 | enemy 0 | ally 0 | ally 3 | memory 0 | self | enemy 3 | memory 0 | enemy 1 |
| 10 | ally 2 | ally 2 | ally 2 | enemy 1 | ally 3 | ally 3 | memory 0 | self | memory 0 | memory 0 | enemy 1 |
| 11 | ally 2 | ally 2 | ally 2 | enemy 0 | ally 2 | ally 3 | memory 0 | self | memory 0 | ally 2 | enemy 3 |
| 12 | ally 0 | ally 0 | ally 2 | enemy 0 | ally 0 | ally 3 | memory 0 | self | memory 0 | self | enemy 5 |
| 13 | ally 2 | ally 2 | ally 2 | enemy 0 | ally 2 | ally 3 | memory 0 | self | memory 0 | self | enemy 5 |
| 14 | ally 0 | ally 0 | ally 1 | enemy 0 | ally 3 | ally 3 | memory 0 | self | memory 0 | self | ally 3 |
| 15 | enemy 0 | ally 0 | enemy 0 | enemy 0 | ally 3 | ally 3 | memory 0 | self | memory 0 | self | ally 3 |
| 16 | enemy 0 | ally 3 | enemy 0 | enemy 0 | ally 3 | ally 3 | memory 0 | self | memory 0 | self | ally 3 |
| 17 | enemy 0 | ally 3 | enemy 0 | enemy 0 | ally 3 | enemy 1 | memory 0 | ally 3 | ally 3 | ally 3 | enemy 1 |
| 18 | enemy 0 | ally 3 | enemy 0 | enemy 0 | ally 3 | enemy 1 | memory 0 | ally 3 | ally 3 | ally 3 | memory 0 |
| 19 | enemy 0 | ally 3 | enemy 0 | enemy 0 | ally 3 | enemy 5 | ally 3 | ally 3 | ally 3 | ally 3 | ally 0 |
| 20 | ally 3 | ally 3 | ally 3 | enemy 0 | ally 3 | enemy 5 | ally 3 | ally 3 | ally 3 | ally 3 | memory 0 |
| 21 | enemy 0 | ally 2 | enemy 0 | enemy 0 | ally 0 | enemy 5 | self | ally 2 | ally 2 | ally 2 | memory 0 |
| 22 | enemy 0 | ally 2 | enemy 0 | enemy 0 | ally 0 | ally 0 | self | self | ally 2 | ally 2 | ally 2 |
| 23 | ally 2 | ally 2 | ally 2 | enemy 0 | ally 0 | ally 0 | self | ally 2 | ally 2 | ally 2 | ally 2 |
| 24 | ally 0 | ally 2 | ally 2 | enemy 0 | ally 0 | ally 0 | self | ally 2 | ally 2 | ally 2 | ally 2 |
| 25 | ally 2 | ally 2 | ally 2 | enemy 0 | ally 0 | ally 0 | self | ally 2 | ally 2 | ally 2 | ally 2 |
| 26 | ally 2 | ally 2 | ally 2 | ally 2 | ally 2 | ally 0 | self | ally 2 | ally 2 | ally 2 | ally 2 |
| 27 | ally 2 | ally 2 | ally 2 | ally 0 | ally 0 | ally 0 | ally 2 | ally 2 | ally 2 | ally 2 | ally 2 |
| 28 | ally 2 | ally 2 | ally 2 | ally 0 | ally 0 | memory 0 | self | ally 2 | ally 2 | ally 2 | ally 2 |
| 29 | memory 0 | memory 0 | memory 0 | memory 0 | memory 0 | self | memory 0 | self | memory 0 | self | ally 2 |

# D Social Impacts

MARL is a powerful paradigm that can model real-world systems, such as autonomous driving and energy distribution. The proposed ATM could accelerate the learning and improve the performance of existing MARL algorithms, thus increasing the practicability of MARL into real-world applications. However, when applying to real-world tasks, the learning process of MARL with ATM still needs some explorations which may lead to unsafe situations. On the other hand, there still exists the risk of using MARL with ATM to do bad things such as using MARL to perform network attacks.

# E   Limitations

Our study may have several limitations in the extreme cases. First, we design our method based on the setting of the partially observable multiagent environments consisting of agents and some factored object entities. For example, if the agents' sight view is too small that few other object entities are in its view, our method may not be applicable. Second, the Entity-Bound Action Layer requires the manual configurations (e.g. the expert experience) of the mapping relationships of each action and its bound entity. If the configuration is not proper, the performance of our method may be affected.