# OpenReview forum: "Transformer-based Working Memory for Multiagent Reinforcement Learning with Action Parsing"
_NeurIPS.cc/2022/Conference — NeurIPS 2022 Accept_

### Official Review · Reviewer_mbJQ · 2022-07-08

**Rating:** 6
**Confidence:** 3
**Soundness:** 3 good
**Presentation:** 3 good
**Contribution:** 3 good

**Summary:**

This paper proposes a new Multiagent Reinforcement Learning (MARL) method using Agent Transformer Memory (ATM). ATM consists of 4 parts; memory, self, allies and entities. Especially, for self, it duplicates the token to use it for Entity-Bound Action (EBA) layer which affects the performance clearly. For memory, ATM uses working memory concept, and other memory types are also evaluated as an ablation study. ATM-based MARL agent outperforms other methods for StarCraft Multi-Agent Challenge (SMAC) and its architecture (EBA and working memory) shows better performance than alternatives (e.g., relational memory or without EBA).

=================================

What happening if they apply appending style memory to their model is interesting and they didn't show that, but except that, they handled lots of concerns through author-reviewer discussion phase. I sustain my score.

**Questions:**

- Could you share the plots in Figure 2 and 6 by comparing in wall-clock time? I think ATM should be slower than GRU, but want to know how much slower. If ATM is not slower than GRU, then it is interesting also.
- Did you try to apply appending style memory or TrXL style memory? Those one can be computationally expensive, but useful from the perspective that it can support to attend the past explicitly.

**Limitations:**

I think that one of the limitations of ATM is computationally more expensive than recurrent module. I don't want to say "so we don't use ATM". I want to discuss the comparison in the aspect of computation or time to get the action from the agents. Even though ATM is more expensive, sometimes we must pick ATM to solve some tasks requiring to infer huge interactions.

**Strengths And Weaknesses:**

Strengths:
- It proposes a new memory module, ATM which can infer the interaction between the agent, allies, entities and memory.
- It evaluates for SMAC, ATM-based agent outperforms GRU-based one which is usually used.
- It reports the ablation studies for EBA and memory types and the number of memory slots.

Weaknesses:
- As shown in [1], the agent with Transformer requires more computations, even though it can benefit for sample efficiency [1,2] or interactions between entities like this paper. For example, in Figure 2 (a) and (b), ATM-based models outperform GRU-based models in steps, but when comparing in wall-clock time, ATM-based model could be much slower than recurrent module-based models.
- The memory following the working memory mechanism is interesting, but it cannot attend the past explicitly. Other transformer-based agents [2,3] get the advantages from attending the past directly, while it cannot do that through ATM architecture.

[1] Parisotto, Emilio, and Ruslan Salakhutdinov. "Efficient transformers in reinforcement learning using actor-learner distillation." arXiv preprint arXiv:2104.01655 (2021).

[2] Parisotto, Emilio, et al. "Stabilizing transformers for reinforcement learning." International conference on machine learning. PMLR, 2020.

[3] Chen, Chang, et al. "Transdreamer: Reinforcement learning with transformer world models." arXiv preprint arXiv:2202.09481 (2022).

---

> ### Author Response · Authors · 2022-08-02
> **Response to Reviewer mbJQ**
>
> We sincerely appreciate the valuable comments from the reviewer. We provide clarification to your questions as below.
>
> [wall-clock time] To make a fair comparison of the wall-clock time. We test ATM-QMIX and GRU-QMIX on smac while ATM-MAA2C and GRU-MAA2C on lbf with the condition that the CPU/GPU/Memory utilization rate keeps less than the maximum load.
>
> We run 200k steps for each method in each map of smac. The results are shown in Table 1 below.
>
> Table 1. Wall-clock time in smac.
> | Map       | ATM-QMIX time |               GRU-QMIX time              |   ATM/GRU ratio |
> |----------------|:----------:|:----------------------------------:|:----------------------------------:|
> | 4m_vs_5m           |  46min |          22min          |              209%
> | 5m_vs_6m     |    45min    |          24min          |               188%
> | 6h_vs_8z     |    44min    |          28min          |                 157%
> | corridor |    57min   |          37min          |                         154%
>
> We run 2000k steps for each method in each scenario of lbf. The results are shown in Table 2 below.
>
> Table 2. Wall-clock time in lbf.
> | Scenario | ATM-MAA2C time |               GRU-MAA2C time              |   ATM/GRU ratio |
> |----------------|:----------:|:----------------------------------:|:----------------------------------:|
> | 3p_vs_5f     |    55min    |          39min          |              141%
> | 4p_vs_5f     |    60min    |          41min          |              146%
> | 4p_vs_6f     |    60min    |          40min          |              150%
>
> We could see that ATM is slower than GRU as ATM needs more computations but their wall-clock time on these tasks is at the same level.
>
> [apply appending style memory or TrXL style memory] If we set the memory slot number to be equal to the past timestep number, ATM could attend the past hidden states explicitly. On the smac and lbf tasks, we found that using the most recent memory (e.g., set memory slot number at 3) could achieve superior performance while increasing the memory slot number does not improve the performance consistently. As we focus on developing a simple yet efficient memory mechanism that considers the multiagent characters, we do not apply the computationally expensive appending style or XL style memory. We will add this discussion in the revised version.

---

> > ### Comment · Reviewer_mbJQ · 2022-08-05
> > **Thank you authors for response for my questions**
> >
> > Hi authors,
> >
> > thank you for addressing the points that I mentioned.
> >
> > [wall-clock time]
> > What I want to see was to compare ATM and GRU-based models in wall-clock time comparison (similar graph with Fig. 2 and 6, but x-axis is relative wall-clock time) to see how much ATM is efficient in wall-clock time. Could you upload that also?
> >
> > As see your tables for wall-clock time comparison and graphs in the paper, ATM looks can solve the tasks better than GRU-based models while it is not much more efficient than GRU in wall-clock time.
> >
> > [apply appending style memory or TrXL style memory]
> > It is interesting. What you mentioned is the far past knowledge over 3 steps is not usually useful for the agent. Then why Transformer-based model could work better than GRU-based models that can encode near past knowledge well? Because it can infer the interactions between entities and allies explicitly?

---

> > > ### Author Response · Authors · 2022-08-08
> > > **Further Response to Reviewer mbJQ**
> > >
> > > Dear Reviewer mbJQ,
> > >
> > > Thank you sincerely for the insightful discussion and below is our response to your further questions and concerns.
> > >
> > > [wall-clock time] Thanks very much for your suggestion, we rerun the 4m_vs_5m in smac and 15x15_3p_5f in lbf, and plot the learning curves in terms of the wall-clock time. We have uploaded the figures (pdf format) in the attachment, which shows that our proposed ATM solve the tasks better than GRU-based models in wall-clock time. Notes that the wall-clock time performance depends heavily on the hardware environment. For example, if we run multiple ATM or GRU instances on a single machine simultaneously and the CPU becomes the bottleneck resource, the wall-clock time would be much longer than running on a single instance. Under this situation, the CPU bottleneck will reduce the ATM/GRU wall-clock time ratio. Thus, we rerun the 4m_vs_5m in smac and 15x15_3p_5f in lbf one instance per time on a single machine respectively for ATM and GRU’s wall-clock time comparison.
> > >
> > > [apply appending style memory or TrXL style memory] ATM’s multi-slot memory mechanism (e.g., 3 slots in smac and lbf or more if necessary) provides the shortcut recurrence to focus on longer sequential information, replacing the GRU’s single path of information flow with a network of shorter self-attention paths. Moreover, as shown in our ablation study in Figure 3(a), ATM indeed benefits from considering the multiagent characters of the factored multiagent observation space and the action space of meaningful entity interactions, and Transformer is the ideal structure to incorporate the multiagent observation/action space characters. Therefore, ATM is a powerful memory structure in the partially observable multiagent environments.

---

> > > > ### Comment · Reviewer_mbJQ · 2022-08-09
> > > > **Thank you for covering my questions**
> > > >
> > > > Hi authors,
> > > >
> > > > thank you for trying to cover my questions.
> > > >
> > > > [wall-clock time] yes, the comparison in wall-clock time is dependent on which hardware is used, but as you mentioned, if same hardware was used, then the comparison should be good enough.
> > > >
> > > > [apply appending style memory or TrXL style memory] yes, Transformer looks suitable architecture to encode the observations from multiagent environments. However, what I want to see was if ATM uses more powerful memory module like appending style (e.g., TrXL or Vanilla Transformer) then the memory capacity will increase which could make better results. If so, it is also interesting, but it can be another project.
> > > >
> > > > Thanks!

---

### Official Review · Reviewer_9Aou · 2022-07-08

**Rating:** 6
**Confidence:** 4
**Soundness:** 3 good
**Presentation:** 3 good
**Contribution:** 3 good

**Summary:**

This paper is about designing a new architecture, Agent Transformer Memory (ATM), for multiagent reinforcement learning where each agent is equipped with its own working memory and action space. Specifically, the authors propose to use a transformer-based architecture to process both the moving entities in the environment and the recent memory slots. The output of the transformer consists of an updated memory slot (to be pushed to the limited-capacity working memory) and the encoded representations for each entity. Then, each agent selects an action (from their respective action space) based on their most recent memory slot and the representations of their surrounding entities. Experiments were conducted on the well-established StarCraft Multi-Agent Challenge (SMAC) and Level-Based Foraging (LBF) environments. The authors test the proposed agent transformer memory with QMIX and QPLEX algorithms commonly used on SMAC, and with MAPPO and MAA2C for experiments on LBF. Empirically, it was shown that agents with memory and personalized action space perform better. The authors also investigate different memory updating schemes and show that using ATM works best.

**Questions:**

- Based on Eq. 4 and 5, should $T_{in}$ and $T_{out}$ be indexed by $i$, i.e., $T_{in}^i$ and $T_{out}^i$?
- In Figure 5, why is Agent 5 heat map incomplete? Was it taken out of action around time step 20?
- From Figure 5, it looks like the memory is not very useful to attend to after the first few couples of steps. Why is that?
- Did the authors try making the trained agents with different architectures play against each other (instead of against the built-in AI)?

**Limitations:**

The authors discussed the limitations and social impacts in the Appendix. Notably, how the learning process requires some exploration that could lead to unsafe situations for both the agents and humans, and also how the Entity-Bound Action Layer requires expert knowledge to manually configure it.


**Strengths And Weaknesses:**

**What I like about this paper**
- Each agent having their own point-of-view of the environment, own working memory, and own action space, while using the same model (same weights, different input $T_{in}$).
- The ablation study of the proposed architecture ATM. It does seem like having a working memory is important (even if only one slot).

**Potential weaknesses**
 - Lack of coordination between the agents. If I understood correctly, all agents predict their next action at the same time. That said, it seems to be a common approach in the MARL literature.
 - Even though this work takes place in a partially observable setting, the proposed technique requires to know in advance how many entities (and their attributes) there will be in the game. Since the proposed architecture requires the relative spatial positions between the self entity and all other entities (even if they are out of the field of view?), it is not clear to me what remains "partially observable".
 - It is not clear how the proposed model performs when playing against other learning agents.

**Originality, quality, clarity, and significance**

As pointed out in the related work, the concept of having a transformer-based working memory is not new. The paper would benefit from stating clearly how the proposed technique differs from each related work. The main thing I could see is the application of a transformer encoder in the multi-agent RL setting with agent-centric observations and working memory. It wasn't clear to me how the Entity-Bound Action Layer related to any previous work (if any). I found the paper technically sound and the ablation study is backing up the proposed agent transformer memory component.

I found the paper well-written and well-organized for the most part. In Figure 1, it is not clear what the gray squares represent, is it the spatial/temporal embeddings, or the entity/memory IDs? While it is unclear how the proposed model compares to the actual state-of-the-art on tested environments, the empirical results suggest the ATM architecture would be useful to others in the community.

Overall, I tend to recommend this paper for acceptance.

**Minor**
- p.2 (l.83): should $d_o$ be $d_e$, if not what is $d_o$?
- p.5 (l.164): "for four times" -> "four times"
- p.7 (l.243): "...agent need remember them." -> "..agent needs to ..."
- p.7 (l.244): "...sight field" -> "..sight range"
- p.7 (l.246): "...focusing fire" -> "..focused fire"

---

> ### Author Response · Authors · 2022-08-02
> **Response to Reviewer 9Aou**
>
> We sincerely appreciate the constructive comments from the reviewer. We provide clarification to your questions and concerns as below.
>
> [Lack of coordination] We follow the setting that all agents predict their next action at the same time, which is common in the MARL literature [1, 2].
>
> [partially observable setting] We use the field of view to introduce partial observability. In detail, if one entity is out of the agent’s view, the information about this entity (including relative spatial positions between this unseeable entity and the agent itself) will not appear in the agent’s local observation. With the help of the transformer, ATM can receive the input of a dynamic agent number or entity number (by setting a maximum entity number and padding zeros to the unseeable entity if this entity is out of view).
>
> [playing against learning agents] Enabling the proposed model to play against other learning agents is a challenging open problem where the environment dynamics are changing with the learning of the opponent agents. It is possible to equip ATM with the opponent modeling methods to explicitly model the opponent learning agents to tackle this problem. This could be an interesting research direction for future work.
>
> [gray squares in Figure 1] The grey squares represent relative positional embeddings with spatial or sequential order information of entities. For memory or duplicated self entities, it is the one-hot ID. For allies or other entities (such as enemies), it is the relative distance between the central self entity and the allies or other entities. We mentioned the grey squares in Figure 1’s caption and explain them in Line 121-128. We will elaborate on our description to make it clearer.
>
> [Question 1] Yes, we will add the indexing as suggested.
>
> [Question 2] It is because that Agent 5 is dead after around time step 20, then the attention mechanism computes Agent 5’s attention weights at near 0.
>
> [Question 3] It is because, after the first few steps of walking, agents are attacking the enemies and walking to avoid being killed. Then the agents are also concentrating on the current allies and enemies to make decisions and give attention weights to these related entities. Appendix C gives a deeper analysis and shows that memory still plays an important role when the agent makes decisions.
>
> [Question 4] Although it is an interesting comparison, the current SMAC platform does not support replacing the built-in AI with trained agents and we follow the setting of previous works such as QMIX and QPLEX.
>
> [Minor] We will revise the minors as suggested.
>
> **Reference**
>
> [1] Lowe, R., WU, Y., Tamar, A., Harb, J., Pieter Abbeel, O., & Mordatch, I. (2017). Multi-Agent Actor-Critic for Mixed Cooperative-Competitive Environments. Proceedings of the 31st Advances in Neural Information Processing Systems, 6379–6390.
> [2] Rashid, T., Samvelyan, M., Witt, C. S. de, Farquhar, G., Foerster, J. N., & Whiteson, S. (2018). QMIX: Monotonic Value Function Factorisation for Deep Multi-Agent Reinforcement Learning. Proceedings of the 35th International Conference on Machine Learning, 4292–4301.

---

> > ### Comment · Reviewer_9Aou · 2022-08-08
> > **Response to authors' rebuttal**
> >
> > Thank you for the insightful rebuttal. All my concerns/questions have been addressed. I still recommend this paper for acceptance.

---

### Official Review · Reviewer_KrWT · 2022-07-10

**Rating:** 6
**Confidence:** 2
**Soundness:** 3 good
**Presentation:** 3 good
**Contribution:** 3 good

**Summary:**

This paper presents an approach for modeling partially-observable multi-agent MDPs. The paper proposes two modifications on top of a general transformer for modeling the relationship between observation and action: first a memory module that maintains information from previous observations associated with all agents, and second a prediction layer that filters information for predicting actions by the arguments of that action (e.g., predicting to attack enemy i is conditioned on the observation information about enemy i). They evaluate on two MARL environments and show improvements over existing methodologies for modeling memory over time, including RNN-based methods.

**Questions:**

* What would alternatives to the entity-bound action layer be? Using only memory of o_self to predict actions, or using some combination (e.g., concatenation) of all observations (which has a drawback of many parameters, some of which may be unnecessary)? I'm a bit confused what it means to ablate EBA in the experiments, and whether you experimented with all three settings (EBA, self-only, and all observations).
* Did you perform any interpretation of the memory? E.g., if memory/observation can be interpreted as an image.

**Limitations:**

The entity-bound action layer seems specific to the action spaces available in the two environments. What if the action space is much lower-level, such that there aren't semantic relationships between a particular action and a particular entity in the environment (e.g., instead of "attack enemy i", an agent may have to move to a strategic position and rotation with multiple navigation actions, then "attack" by firing some weapon in the direction it is pointed)? Would the entity-bound action layer still generalize to this?

**Strengths And Weaknesses:**

Strengths:

This is an interesting problem and an interesting approach to the problem. The paper is relatively clear and experiments on both environments are mostly comprehensive and performance over compared methods is impressive.

---

Weaknesses:

The way that transformer-based working memory is introduced in 42-48 makes it seem as though the main contribution is applying an existing method to an existing problem. I'm not familiar enough with related work to make this judgment, however.

I was confused about what an observation includes: is o_ally the observation *of* an ally, or the ally's observation? If the latter, does this also hold for enemies, and if so, it seems as though an agent shouldn't have access to the enemy's observation. If the former, is there information sharing between agents (i.e., sharing their first-person observations, which may not overlap with another ally's)? How would this affect performance? If would help to have a formal definition of the task, including what exactly observations look like (are they images?).

---

> ### Author Response · Authors · 2022-08-02
> **Response to Reviewer KrWT**
>
> We sincerely appreciate the insightful comments from the reviewer. We provide clarification to your questions and concerns as below.
>
> [transformer-based working memory] While previous works focus on the single-agent memory mechanisms, we propose a unique transformer-based multiagent memory mechanism by considering the multiagent characters in both the observation space and action space. In particular, for the first time, we develop a novel memory structure with the help of the working memory updating mechanism by explicitly considering the allied agents for the factorized multiagent observation space. We will elaborate on our description of this contribution.
>
> [observation] The $o_{ally}$ is the agent’s observation of an ally such as the ally’s attribution features (including the ally’s health and relative coordinates between the ally and the agent). Agents cannot share information such as their first-person observations among themselves as we assume there is no communication. At the same time, if one entity is in both agent $i$ and agent $j$’s sight ranges, then agents $i$ and agent $j$’s own local observations include the same seen entity. In Line 186-192, we provide the details of each entity type (vector features instead of images) in the smac tasks as an example. In smac, the exact observation of agent $i$ is a set of entities $e$ where $e$ ∈ {$o_{self}^{i}$,**o**$^{i}\_{ally}$,**o**$^{i}\_{enemy}$} and $d(e,i)<d_{sight}^{i}$ ($d(e,i)$ is the distance between the entity $e$ and agent $i$, and $d_{sight}^{i}$ is agent $i$’s sight range). We are glad to elaborate on our description of the observation as suggested.
>
> [alternatives to the entity-bound action layer] As stated in Eq. (7) and Line 148-150, one common alternative to the EBA layer is using linear layers to map the self entity embedding to all action nodes’ values. Here we use this setting to ablate EBA in the experiments as the self entity embedding also contains the information from all entities after the transformer. We will elaborate on our description of the ablation setting.
>
> [interpretation of the memory] In Figure 5(a), the memory attention heatmap shows how the newly generated memory focuses on each kind of entity during the whole episode. In Figure 5(a), the enemy and ally entities receive the most attention weights after the first few couples of steps, which means the enemy and ally information is absorbed into the new generated memory slot during these timesteps. This helps interpret what the memory consists of.
>
> [Limitation] It is interesting and possible to generalize the entity-bound action layer into the lower-level action space. However, the low-level actions such as rotation or firing in a certain direction may be related to different entities at different timesteps, which is difficult to configure manually or by rule. For this purpose, it needs to design a mechanism to automatically map entities to actions, which is not trivial. We list it as our future work as shown in the conclusion section.

---

> > ### Comment · Reviewer_KrWT · 2022-08-09
> > **Thanks**
> >
> > Thank you for the response. I have no remaining concerns and still suggest to accept the paper.

---

### Meta-Review · Area_Chair_jXHt · 2022-08-29

**Recommendation:** Accept
**Confidence:** Certain

**Metareview:**

All reviewers agree that this paper makes a good contribution in developing a novel transformer-based memory structure for MARL. The developed approach is evaluated through comprehensive and solid experiments. The authors have also clearly addressed the questions/concerns raised by the reviewers.

**Award:**

No

---

### Decision · Program_Chairs · 2022-09-14

Accept